# Surface Cleaning and Passivation Technologies for the Fabrication of High-Efficiency Silicon Heterojunction Solar Cells

**DOI:** 10.3390/ma16083144

**Published:** 2023-04-16

**Authors:** Cuihua Shi, Jiajian Shi, Zisheng Guan, Jia Ge

**Affiliations:** College of Materials Science and Engineering, Nanjing Tech University, Nanjing 211816, China

**Keywords:** heterojunction, high efficiency, surface cleaning, passivation, carrier selectivity, thin wafer, SHJ solar cell

## Abstract

Silicon heterojunction (SHJ) solar cells are increasingly attracting attention due to their low-temperature processing, lean steps, significant temperature coefficient, and their high bifacial capability. The high efficiency and thin wafer nature of SHJ solar cells make them ideal for use as high-efficiency solar cells. However, the complicated nature of the passivation layer and prior cleaning render a well-passivated surface difficult to achieve. In this study, developments and the classification of surface defect removal and passivation technologies are explored. Further, surface cleaning and passivation technologies of high-efficiency SHJ solar cells within the last five years are reviewed and summarized.

## 1. Introduction

Motivated by the escalating severity of ecological damage and the gradual depletion of traditional energy sources, the utilization of renewable energy sources is gaining momentum, particularly in light of the global consensus to achieve carbon neutrality. The annual increase in renewable energy capacity broke a new record in 2021, with an increase of 6% to nearly 295 GW [1]. Solar power has the advantages of abundance, safety, and cleanliness, with solar cells being able to convert the energy of light into electricity via the photovoltaic (PV) effect [2]. In 2021, solar photovoltaics alone accounted for over half of all renewable energy expansion, followed by wind and hydropower [3]. According to the International Energy Agency, global solar PV capacity is forecast to nearly triple between 2022–2027, surpassing coal to become the world’s largest source of electricity capacity [4]. Due to its abundant material availability, relatively low cost, and well-established industrialization technology, silicon wafer solar cells comprise over 90% of the global PV market [5]. Passivated Emitter and Rear Cell (PERC) is the current mainstream technology, which faces the problem of efficiency improvement. At present, the efficiency of PERC in mass production has reached 23.3%, which is considerably close to the practical efficiency of 24.5% [6]. Passivated contact (PC) technology is indispensable in creating efficiency record-level solar cells [7,8,9,10]. A promising PC technology is heterojunction contact, which involves the use of a hydrogenated amorphous silicon (a-Si:H) layer to passivate the crystalline silicon (c-Si) substrate. From the technical report of the National Renewable Energy Laboratory (NREL) [11], the wafer constitutes the most significant portion of the total cell cost. Cost reduction drives wafers to become increasingly thinner, which, in turn, imposes higher requirements for subsequent processes. According to Infolink in 2022 [12], most of the current SHJ cells are based on 165 µm thick wafers. Leading companies currently use 130 μm thick wafers for cell production, with plans to gradually reduce wafer thickness [13]. However, the optimal levelized cost of electricity (LCOE) occurs when the wafer thickness is around 50 µm thick, which can reduce manufacturing costs and capital expenditure (CAPEX) by 48%, module costs by 28%, and ultimately, LCOE by 24% [14]. Regarding the structure of intrinsic amorphous silicon and phosphorus-doped amorphous silicon double-sided passivated silicon wafers (a-Si:H(n)/a-Si:H(i)/c-Si(n)/a-Si:H(i)/a-Si:H(n)), the implied efficiency was observed to reach a maximum of 28.4% at a wafer thickness of approximately 100 μm thick [15]. Open-circuit voltages (V_OC_) over 760 mV have also been obtained on SHJ structures based on 50 µm thick [16] and 40 µm thick [17] wafers, demonstrating the high passivation quality by using ultra-thin silicon wafers in SHJ structures. The aforementioned laboratory and industrial results have revealed the notable potential of SHJ solar cells when employing ultra-thin wafers (<100 µm) while maintaining salient performance.

SHJ cells show significant benefits and potential for industrialization but are restricted by the high cost of n-type wafers and deposition equipment [13,18], making the high-efficiency cells and the use of low-cost cleaning methods highly desirable. High power conversion efficiency (PCE) is highly related to high V_OC_, which is strongly dependent on the quality of interface passivation [9,19]. The complex nature of the passivation layer and preceding cleaning procedures renders the achievement of a well-passivated surface challenging. Thus far, several reviews on passivation contact have been reported. Melskens et al. [20] reviewed the use of passivated contacts in c-Si solar cells, particularly for dopant-free materials. Panigrahi and Komarala [21] summarized the research progress in the preparation of a-Si:H(i) films for passivating c-Si using a plasma enhanced chemical vapor deposition (PECVD) system. Sun et al. [22] reviewed the research development of SHJ solar cells from the perspectives of V_OC_, short-circuit current density (J_SC_), and fill factor (FF). Muduli and Kale [23] reviewed the evolution of different passivated structures in c-Si solar cells and discussed passivation strategies for the currently studied mainstream structures. To our knowledge, few reviews on PV surface cleaning have been reported. In this study, the progress made in silicon heterojunction solar cells, cleaning methods, and surface passivation technologies are reviewed, and the wafer cleaning and surface passivation technologies employed in high-efficiency SHJ solar cells over the past five years are summarized.

## 2. Overview

### 2.1. SHJ

Heterojunction was first proposed in 1948 [24]. In 1974, Fuhs et al. [25] first proposed the preparation of heterojunction structures through the deposition of amorphous silicon onto silicon single crystals. The most important feature is that the advantages of crystalline silicon and thin film cells were combined. Sanyo (now acquired by Panasonic) was a pioneer in applying the a-Si:H(i) thin film to a-Si:H/c-Si heterojunction solar cells [26]. The insertion of an a-Si:H(i) layer greatly reduces the recombination of interfacial carriers and thus improves the PCE [27]. The schematic structure of a SHJ solar cell is shown in Figure 1a. After texturing and cleaning, a-Si:H(i) layers are deposited on both sides of the silicon wafer. Boron-doped amorphous silicon (a-Si:H(p)) thin film is then deposited on the front or back side to form the emitter, which creates a p-n junction with the substrate to induce band bending, thereby enabling charge separation [28]. a-Si:H(n) thin film is deposited on the other side of the wafer to form a back or front surface field (BSF or FSF), which serves to repel minority carriers, thereby reducing carrier recombination [28]. As shown in the band diagram (Figure 1b), band offsets are present at each a-Si:H/c-Si interface [29]. The discontinuity at the conduction band edge (E_C_) is equivalent to the difference in electron affinity (χ_c-Si_ = 4.05 eV and χ_a-Si:H_ = 3.9 eV), whereas the discontinuity at the valence band edge (E_V_) is contingent upon the difference in electron affinity and band gap [28]. As a result, there is a large band offset at the valence band edge (~0.36 eV) and a small band offset at the conduction band edge (~0.15 eV) [30]. After the deposition of the doped amorphous film, transparent conductive oxide (TCO) layers are deposited on each side as the anti-reflection layer and lateral conducting layer. Finally, electrode fabrication methods, such as screen printing, in which low-temperature curing silver pastes are usually printed, are applied. Due to the high bifacial capability (up to 95%) [11,31], simple and low-temperature process [32,33,34], and superior temperature coefficient [35], SHJ solar cells continue to attract interest in the field of high-efficiency solar cells.

### 2.2. Surface Cleaning Techniques

#### 2.2.1. Radio Corporation of America (RCA) Standard Cleaning

In 1965, ‘RCA Standard Cleaning’ was developed and initially consisted of simple immersion in hot alkaline and acidic hydrogen peroxide solutions at 75 to 80 °C for 10 min [36]. The RCA Standard Cleaning contains two cleaning steps, namely, RCA1 (or SC1) and RCA2 (or SC2). The molecular mechanism can be found in Ref. [37]. The first cleaning step (RCA1) involves the use of an alkaline mixture to remove organic contaminants and remove certain Group IB and IIB metals. The common volume ratio of the mixture is ammonium hydroxide (NH_3_∙H_2_O): hydrogen peroxide (H_2_O_2_): H_2_O = 1:1:5 [36]. The second cleaning step (RCA2) involves the use of an acidic mixture for the removal of heavy metal ions, commonly in the ratio of hydrochloric acid (HCl): H_2_O_2_:H_2_O = 1:1:6 [36]. Improvements to the RCA cleaning technique, initially developed for integrated circuits [36], have already been implemented in solar cell manufacturing. However, the large amount of cleaning steps and the significant chemical usage in the commonly employed RCA cleaning technique (RCA1, rinsing, diluted hydrofluoric acid (DHF), rinsing; RCA2, rinsing, DHF, rinsing, and drying) render it impractical for mass production [38]. Simplified variations of such cleaning procedures were investigated to fulfill the demands of cleaning efficiency and industrial output [39,40]. For surface cleaning and smoothing, a modified RCA1 followed by a HCl/HF step was established [40]. The modified RCA1 consists of low concentration H_2_O_2_ and potassium hydroxide (KOH) or sodium hydroxide (NaOH), which reduces the cost. DHF dip is often used after RCA2 cleaning to remove the surface oxide layer and make the surface hydrophobic, acting as chemical passivation [41]. Strinitz et al. [40] used the same HCl/HF solution to eliminate metal contamination and oxide layers, thereby reducing the duration of the cleaning process. By assessing the cleaning impact of each cleaning stage, Liu et al. [42] achieved a cleaning procedure with effective surface passivation and a streamlined cleaning process. Reducing the cleaning duration of RCA1 to 1/2, omitting the RCA2 step, and cutting the wet-chemical oxidation time resulted in a superior cleaning effect compared to the original scheme. The cleaning time of RCA1 was reduced to 1/4, which reduced the duration from 90 min to 25 min, achieving a cleaning effect comparable to the original scheme.

With regard to the environmental and economic aspects, the PV community has been continuously exploring alternative cleaning methods to replace the wet-chemical approach.

#### 2.2.2. Ozone-Based Cleaning

Ozone (O_3_) dissolved in deionized water possesses strong oxidizing properties and can be utilized alone to eliminate organic contaminants [43]. A mixture of ozone and acidic solvent (for example, HF and/or HCl) is effective at removing organic and metallic contaminations [44,45]. Figure 2 shows an ozone-based cleaning system, in which ozone is generated by a generator and mixed with acidic solvent in the static mixer [46]. The mixed O_3_/HF/HCl solution is then fed into the process bath through a recirculation pump. The temperature of the solution is monitored by a temperature detector in a process bath, while the ozone concentration is regulated by an ozone sensor.

Ozone-based cleaning has been extensively studied as a substitute for certain wet-chemical cleaning processes, enabling the production of high-efficiency solar cells with reduced costs and minimized environmental pollution. Sonner et al. [47] discovered that among KOH, RCA1, and HCl/O_3_ cleaning procedures, HCl/O_3_ exhibited the optimal removal ability for metal contaminations and equivalent removal efficiency for organic contaminants. Chemical passivation was achieved by employing a DHF dip at the end of the process, resulting in a hydrogen-terminated interface. Kranz et al. [48] reported that the HF/O_3_ cleaning method could produce a hydrophobic surface that was similar to the RCA method. Li et al. [49] reported that the use of room temperature O_3_/HF/HCl for 10 min achieved a higher passivation quality than the RCA procedure, with an effective carrier lifetime (τ_eff_) of 5.8 ms. A τ_eff_ of 7.3 ms was obtained on a structure that was passivated using a 20 nm thick intrinsic layer on both sides (a-Si:H(i)/c-Si(n)/a-Si:H(i)), indicating an implied open-circuit voltage (i-V_OC_) of 740 mV at 1 sun illumination [40].

H_2_O_2_ is involved in pre- and post-texturing cleaning and accounts for about 50% of the total cost of typical chemicals [38]. As shown in Figure 3, H_2_O_2_ in the RCA cleaning process can be avoided by using ozone-based cleaning. By using O_3_ in the pre-cleaning step and O_3_/HCl/HF in the post-cleaning to remove organic and metal contaminants, the consumption of chemicals is greatly reduced and the environmental pollution from chemicals is mitigated. [50]. Compared with traditional RCA wet-chemical cleaning, the currently widely used ozone cleaning method can reduce the cost by 13%, and further optimization of the ozone cleaning process can reduce the cost by 30% [38].

Ozone-based cleaning is regarded as a compelling substitute for RCA cleaning due to its potential to effectively eliminate surface contaminants, decrease the requirement for chemicals, lower wastewater treatment costs, and reduce cleaning time. However, to further develop such a cleaning method, stability and repeatability issues need to be addressed [51].

#### 2.2.3. Plasma Etching (PE)

Passivation achieved by the final DHF dip is considered meta-stable due to the gradual replacement of weak silicon-hydrogen bonds by oxygen–silicon bonds when wafer is exposed to air for a few seconds [52]. Dry cleaning methods have been investigated for etching the native oxide layer while avoiding reoxidation. As shown in Figure 3, after RCA cleaning process, silicon wafer is placed directly into the PECVD chamber to remove the native oxide layer from the silicon surface. Moreno et al. [53] reported a two-step process (410 s) using silicon tetrafluoride (SiF_4_) followed by brief hydrogen plasma exposure to replace HF cleaning, resulting in a τ_eff_ up to 1.55 ms. In addition, a hydrogen plasma-based one-step process has been developed as an alternative to the HF cleaning method [54]. The active species in the plasma react with the surface oxide layer to form volatile products [55], which are then pumped out of the chamber. Tang et al. [56] used short-time (20 s) hydrogen plasma cleaning to remove the oxide layer, achieving a τ_eff_ value of 3.7 ms after thermal annealing, which is comparable to conventional HF cleaning. Xu et al. [57] also confirmed that utilizing hydrogen plasma treatment prior to a-Si:H(i) deposition can serve as a feasible substitute to HF cleaning. Furthermore, surface after H_2_ plasma etching is highly hydrogenated, which prevents epitaxial growth during a-Si:H deposition and thermal annealing, and a τ_eff_ of 2.5 ms is obtained on sample that is double-side coated with 12 nm thick a-Si:H(i).

A significant advantage of such dry etching in manufacturing is that both surface cleaning (the last step) and deposition can be performed in the same system without interrupting the vacuum, thereby avoiding the risk of re-oxidation and recontamination [58,59]. However, the optimal trade-off between effective plasma cleaning treatment and preservation of undamaged silicon surfaces requires further investigation [57].

### 2.3. Surface Passivation

#### 2.3.1. Chemical Passivation

Surface chemical passivation can be accomplished through immersion of wafer in a polar solution to minimize interfacial states or by saturating silicon dangling bonds using passivation layer(s). Typically, both methods are used simultaneously, which allows sufficient surface chemical passivation to completely saturate the dangling bonds, making it ideal to prevent carrier recombination for both electrons and holes [60].

After RCA cleaning, a brief immersion in DHF solution is commonly used to remove the surface oxide layer [61] and passivate surface defects [41]. The principle of surface passivation has evolved from F termination [62] to the more effective H termination [63,64]. As early as 1986, the surface of silicon cleaned with HF after oxidation was reported to be covered with covalent Si-H bonds and there were almost no surface dangling bonds remaining [64]. Intrinsic amorphous silicon films are typically deposited as a chemical passivation layer following HF cleaning, so as to lower the defect density induced by the direct deposition of doped amorphous silicon films on c-Si [26]. Films deposited in the transition zone from amorphous to the crystalline phase have been demonstrated to exhibit high surface passivation quality [65,66]. At this point, the a-Si:H(i)/c-Si interface is completely relaxed, and the number of electron-active defects is minimal, without an epitaxial layer [67]. The passivation quality provided by the a-Si:H(i) layer typically decreases with the deposition of the a-Si:H(p) layer on top of it [68], with the passivation degradation being potentially attributed to Si-H fracture in a-Si:H(i) film [69]. Liu et al. [70] reported that the sample in the transition zone had a τ_eff_ of 807 µs for structures passivated by a-Si:H(i) on both sides, which was substantially higher than the 8 µs of the less dense sample. The SHJ cell with less dense a-Si:H(i), on the other hand, displayed higher current-voltage (I-V) performance. Passivation degradation of sample in cell has been related to the epitaxial growth that occurs after doped thin film deposition. Since less dense a-Si:H(i) samples contain a high number of voids (for example, Si-H_2_), their mass density is lower than that of the transition zone sample. Such porosity of the a-Si:H(i) layer allows H to penetrate into the amorphous network and the a-Si:H(i)/c-Si interface after the deposition of the doped layer, preventing the formation of the epitaxial layer [70]. According to Wu et al. [71], the compactness of films deposited at R_H_ (H_2_/(H_2_ + SiH_4_)) of 0 and 3 was lower than that at R_H_ = 0.45. Higher τ_eff_ values were obtained in the passivated structure (a-Si:H(n)/a-Si:H(i)/c-Si(n)/a-Si:H(i)/a-Si:H(p)) at R_H_ of 0 and 3. In another study, it was found that when a less dense a-Si:H(i) layer was deposited on both sides of a 10 nm thick a-Si:H(n^+^) layer, the τ_eff_ increased to 19.9 ms, which was significantly higher than the τ_eff_ of 15.2 ms obtained when using a dense a-Si:H(i) film [72]. Therefore, the less dense a-Si:H(i) has a higher level of hydrogenation at the a-Si:H(i)/c-Si(n) interface than the dense a-Si:H(i), which can facilitate hydrogen diffusion in the subsequent annealing and deposition of doped amorphous films. The SHJ cell prepared using less dense intrinsic layer and 8 nm a-Si:H(n^+^) achieved an efficiency up to 23.0% [72]. Cells prepared from less dense a-Si:H(i) had a higher V_OC_ (741 mV) than cells prepared from the dense a-Si:H(i) film, indicating the advantage of the less dense a-Si:H(i) for high passivation quality in cell preparation [72]. Despite such results, the porous structure was reported to widen the band gap of the a-Si:H(i), thereby increasing ∆E_V_ [71]. Consequently, the porous and less dense a-Si:H(i) was found to be unsuitable for passivation at the emitter side of n-type SHJ solar cells. The structure of the less dense a-Si:H i1 layer (deposited by pure silane) capped by a dense a-Si:H i2 layer (deposited by silane diluted with H_2_) has been shown to improve interface passivation quality [73,74]. Due to the ultra-thin i1 layer (1–5 nm), hydrogen can penetrate this void-rich layer and passivate defects at the a-Si:H(i)/c-Si interface [74]. As depicted in Figure 4, the i1 layer has a highly porous structure that can prevent epitaxial growth [75]. The i2 layer has low defects and low porosity, which results in a denser film and, therefore, provides good passivation quality [76,77]. The insertion of an i1 buffer layer on both sides of a SHJ solar cell resulted in a high conversion efficiency of 23.5%, with a V_OC_ of 740 mV, a J_SC_ of 38.7 mA/cm^2^ and a FF of 82% [78]. However, high hydrogen content may lead to excessive defects and voids, resulting in increased ∆E_V_, thereby decreasing FF and PCE.

Hydrogenated amorphous silicon carbide (a-SiC_x_:H(i)) exhibits good temperature stability and a larger band gap than a-Si:H(i) [79]. The optical band gap of a-SiC_x_:H(i) can be expanded to 4 eV at high methane concentrations [79], which reduces optical absorption loss in the passivation layer [80]. However, at the a-SiC_x_:H(i)/c-Si interface, high interface trap density was observed with high carbon atomic concentrations [81]. Ehling et al. [82] demonstrated that a low carbon content ([CH_4_]/[(CH_4_) + (SiH_4_)]) of 1.3% was associated with a higher lifetime value. In the case of low carbon incorporation, a low absorption strength ratio indicates a dense structure; but as the carbon content increases, the diffusion of hydrogen to the interface becomes blocked due to the stronger C-H bond. The resulting decrease in hydrogen atoms at the a-SiC:H(i)/c-Si interface leads to a decrease in the passivation quality [82]. Donercark et al. [81] proposed the use of a stacked structure using a-Si:H(i) and a-SiC_x_:H(i) to improve the quality of interface passivation while ensuring that the large band gap a-SiC_x_:H(i) was leveraged to reduce optical absorption loss. The incorporation of carbon into a-SiC:H results in the formation of structural defects and inhomogeneities, leading to disorder in amorphous networks. Meanwhile, a-SiO:H, which has a higher electronegativity than a-SiC:H, is capable of producing a lower defect state [83]. In previous research, the tunable band gap [84] and high-quality surface passivation [33,85] have made a-SiO:H(i) a viable material for the passivation layer. Since O is substantially more electronegative than Si, the incorporation of oxygen atoms caused surrounding H atoms to become more strongly attracted, resulting in additional Si-H_2_ bonding configurations and porous structures [86]. Therefore, the incorporation of oxygen atoms enhances the porosity of the intrinsic film, allowing hydrogen to penetrate more easily into the film and passivate interface dangling bonds. The use of a-SiO:H as passivation material has been shown to be effective in inhibiting epitaxial growth. Such findings can be attributed to the abrupt interface formed by the a-SiO:H(i) passivation layer, whereas the same gas ratio of R_H_ ([H_2_]/[SiH_4_]) results in epitaxy of the a-Si:H(i) passivation layer [86]. The higher J_SC_ observed in a-SiO_x_:H(i) passivated cells compared to less dense a-Si:H(i) passivated cells can be attributed to the large band gap of a-SiO_x_:H(i), which reduces parasitic absorption loss in SHJ solar cells. When deposited on the front side, the a-SiO_x_:H(i) passivation layer improved J_SC_, while a-Si:H(i) provided a good post-passivation interface as a passivation layer on the rear side, resulting in a PCE of 22.2% [86]. When using a large band gap a-SiO_x_:H(i) passivation layer, the balance between passivation and carrier transport must be evaluated. Zhang et al. [87] reported that as the Fo ([CO_2_]/[CO_2_ + SiH_4_]) increased from 0.33 to 0.4, the increase in ∆E_V_ favored photogenerated electron transport, as it compensated for the absorption loss in the a-SiO_x_:H(i) passivation layer due to porosity and high defect density. Fo above 0.4 results in lower I-V performance due to poor passivation quality on the backside and blocked hole transport caused by a higher energy barrier of ∆E_V_. To prevent hole transport barriers caused by high valence band mismatch, it is recommended that the a-SiO_x_:H(i) layer be used as a passivation layer on the FSF side of the n-type silicon wafer in SHJ solar cells [84,86].

#### 2.3.2. Field-Effect Passivation

Field-effect passivation is achieved by breaking the balance of carrier concentrations at the interface [28]. The presence of the built-in potential can induce band bending and provide an energy barrier, leading to the accumulation and depletion of either carrier type, thus reducing carrier recombination. In traditional SHJ solar cells, n-type and p-type a-Si:H are used to provide field-effect passivation by forming accumulation and inversion regions at the c-Si(n)/a-Si:H(i)/a-Si:H(n) and c-Si(n)/a-Si:H(i)/a-Si:H(p) interfaces, respectively [18]. Thus, asymmetric distribution of electrons and holes is formed at the respective contact interfaces, leading to carrier selectivity [18]. In the context of SHJ solar cells, n-type carrier-selective materials enable the transport of electrons while preventing the transport of holes, whereas p-type carrier-selective materials perform the reverse function. Such materials should be sufficiently doped to support carrier separation and collection [88]. However, due to the relatively narrow band gap and the high defect density within the doped a-Si:H, undesired parasitic optical absorption occurs in the ultraviolet and visible ranges of the solar spectrum [89]. Considering the low doping efficiency of a-Si:H [88], two main materials have been investigated, namely, Si-based materials and dopant-free materials.

(1)Si-based materials:

Hydrogenated micro/nanocrystalline Si-based materials, such as hydrogenated micro/nanocrystalline silicon (μc/nc-Si:H), hydrogenated micro/nanocrystalline silicon carbide (μc/nc-SiC_x_:H) and hydrogenated micro/nanocrystalline silicon oxide (μc/nc-SiO_x_:H), with high doping efficiency [88] and large band gap, have been investigated as carrier-selective materials. The terms nanocrystalline and microcrystalline are mainly used to describe materials with different grain sizes and crystallization fractions [90]. Unless specifically indicated in the literature, nanocrystalline (nc) in the following text is the collective term for nanocrystalline and microcrystalline.

Hydrogenated nanocrystalline Si-based materials consist of both amorphous and nanostructured phases [91,92,93]. In principle, nc-Si:H, nc-SiO_x_:H, and nc-SiC_x_:H can be treated as nc-Si:H embedded in a-Si:H, a-SiO:H, and a-SiC:H, respectively [94,95,96]. The alloying features of nc-SiC_x_:H and nc-SiO_x_:H result in their large band gaps, which can reduce optical absorption. nc-SiC:H(n) exhibits a low absorption coefficient for photon energies greater than 1.75 eV [97], indicating its transparency that results in low absorption loss. Köhler et al. [98] reported that using hot wire chemical vapor deposition (HWCVD) to deposit μc-SiC:H(n) at higher filament temperature (T_f_) results in a decrease in i-V_OC_, which is due to the increased hydrogen density at high T_f_ (1850 °C), resulting in the SiO_2_ passivation layer being etched. A similar phenomenon occurs when µc-SiC:H(n) and µc-SiO:H(n) are used as electron-selective layers (ESL) [99]. This is due to higher T_f_, causing hydrogen in precursor gas to diffuse through microcrystalline and amorphous silicon thin films and gradually form bubbles in the micropores at the interface of a-SiO_x_:H(i)/c-Si(p), thereby reducing the passivation quality [99]. However, high T_f_ is of importance to high conductivity. With the increase in T_f_ (1700–1900 °C), the conductivity increased by nine orders of magnitude [98]. The conductivity of μc-SiC:H(n) is provided by the nanostructured phase [100]. A much higher electrical conductivity of nc-Si:H than that of the a-Si:H layer can be achieved with the addition of only a few parts per million (ppm) of doping gas [101]. As a result, nc-SiC_x_:H and nc-SiO_x_:H have higher conductivity and more significant field-effect passivation than a-Si:H [88,101]. However, due to the inherent nature of the crystal phase, such high conductivity can only be achieved after the nucleation stage. Microcrystalline nucleation requires poorly connected silicon networks to reduce the energy barrier to transition from a-Si:H to nc-Si:H under high-hydrogen dilution atmosphere [102]. For high-efficiency cells, it is imperative to deposit highly conductive (namely sufficiently high crystallization) nanolayers on a thin intrinsic passivation layer and ensure excellent passivation quality of the intrinsic layer [103]. One possible solution is to deposit a seed layer after the intrinsic layer and before the transport layer to rapidly facilitate initial nucleation [103,104,105]. Mazzarella et al. [103] reported that a higher J_SC_ without a seed layer (µc-Si:H) was countered by the decrease in V_OC_ and FF, resulting in decreased efficiency. Finally, by depositing a seed layer of µc-Si:H before the µc-SiO_x_:H film, a high cell performance of 22.6% was obtained. Pham et al. [104] also demonstrated that a µc-SiO_x_:H film with a seed layer of µc-Si:H can trigger a rapid transition from amorphous to nanocrystalline phase, causing increased efficiency.

Although the advantages of high doping efficiency and large band gap of hydrogenated nanocrystalline silicon materials were widely studied, the trade-off between optical and electrical properties remains challenging due to the nature of the two-phase structures.

(2)Dopant-free materials:

Dopant-free materials usually have large band gaps and are capable of providing carrier selective contacts, enabling the separation of electrons and holes. Such separation can be achieved by utilizing materials with sufficiently high or low work functions (WF) on either side of a crystalline silicon wafer [106]. Hole selective layer (HSL) requires a WF higher than 5.5 eV, while ESL requires a WF lower than 4 eV [107].

Numerous studies have been reported in the literature on the use of dopant-free materials to achieve hole selective contact. Some examples of such materials include graphene [108], poly(3,4-ethylene dioxythiophene):poly(styrene sulfonate) (PEDOT:PSS) [109] and transition metal oxides (TMOs) [110,111,112,113]. Among such materials, large band gap TMOs are particularly preferable for SHJ solar cells because of the following advantages: (1) cheap, no doping issues and low-cost deposition methods; and (2) less material required because of the smaller thickness [60]. Due to the nature of large band gap (3–3.3 eV) and high WF (4.5 eV–6.9 eV) [114], MoO_x_ (2 < x < 3) has been extensively investigated for SHJ solar cells as an alternative to p-type a-Si:H [115,116,117]. The use of large bandgap MoO_x_ avoids parasitic absorption loss and defect densities in a-Si:H(p) [118], thus reaching a notable J_SC_ value of 39.4 mA/cm^2^ with little optimization required [117]. The high WF of MoO_x_ was reported to improve the hole selective property because of the enhanced built-in potential [119]. The high WF of MoO_3_ results in its Fermi energy level being in close proximity to the valence band of c-Si. This leads to the formation of a large ∆E_C_ at the MoO_x_/c-Si(n) interface, which, in turn, induces a hole inversion layer [120,121]. This inversion layer enables the transport of holes from the c-Si layer to the MoO_x_ layer, while hindering the transport of electrons. Such an effect has been demonstrated in simulation results, where tuning the WF from 4.5 eV to 5.7 eV resulted in an increase in efficiency from 1.62% to 23.32% [122]. MoO_x_ is a TMO typically dominated by oxygen vacancy defects, and the WF of the film decreases as the oxygen vacancy concentration increases. The increase in oxygen vacancy concentration is accompanied by a decrease in the cation oxidation state (Mo^+6^→Mo^+5^→Mo^+4^), from an insulator (MoO_3_) with a high WF to a semiconductor (MoO_3−x_) and then to a metal-like conductor (MoO_2_) with a low WF [123,124]. The decrease in WF due to oxygen vacancy can be attributed to the formation of cations with lower electronegativity and an increase in the density of occupied states [125]. Increasing oxygen vacancies are accompanied by higher levels of low-valence Mo cation, which has low electronegativity. As shown in Figure 5, oxygen vacancies are given donor states near the Fermi level (E_F,d_). The increase in oxygen vacancy concentration causes the donor state near E_C_ to ionize. This leads to an increase in the carrier density in the conduction band and an upward shift in the E_F,d_ to the vacuum level (E_vac_), resulting in the decrease of work function [125]. Compared to the decrease in work function caused by donor state, the formation of low electronegative cations is a major contributor to the reduction of work function. As indicated in Figure 6, when oxygen vacancy density is low, the work function of MoO_x_ declines quickly; but as oxygen vacancy density rises, the drop in work function tends to be gentle [125]. Annealing under an O_2_ atmosphere is an effective method for controlling oxygen vacancy and inhibiting the formation of Mo^5+^ in MoO_x_ films. As shown in Figure 7, annealing under an oxidizing atmosphere of oxygen (O_2_) results in films with a higher Mo^6+^ state and a lower Mo^5+^ state, which is accompanied by a decrease in oxygen vacancies [123]. Annealing under a nitrogen (N_2_)-reducing environment leads to the formation of low-valence Mo^5+^ cations, and the strength of the Mo^5+^ state increases significantly with the increase of annealing temperature. After annealing at 500 °C, there are low electronegativity cation Mo^4+^ peaks, which led to a more significant decrease of work function. On the other hand, high temperatures provide the phase change energy for the crystallization of the film. The β-MoO_x_ phase initially present undergoes gradual crystallization to form the α-MoO_x_ phase [126,127,128]. Compared with β-MoOx, α-MoO_x_ causes a higher degree of lattice distortion on the silicon surface, which results in a higher surface density, indicating poor interface passivation [129].

Low WF TMOs [20], alkali and alkaline earth metal salts [60] have been used as electron transport layers in SHJ solar cells. Within this group, alkali metal fluorides (AMFs) such as lithium fluoride (LiF_x_) and rubidium fluoride (RbF_x_) have attracted particular interest. LiF_x_ exhibits a low WF of about 2.86 eV, enabling the transport of electrons and the blocking of holes in the context of n-type wafer, thereby reducing the recombination rate at the interface and achieving electron-selective contact [106]. The Tauc energy gap of the LiF_x_ is greater than the measurement range (>6.8 eV), resulting in negligible absorption [106]. Singh et al. [130] simulated the back-contact by utilizing LiF_x_ as the ESL with an Ag/Indium Tin Oxide (ITO)/MoO_x_/c-Si(n)/LiF_x_/Al structure. The cell containing LiF_x_ shows a downward band bending, which increases the diffusion length of minority carriers (holes), resulting in enhanced field effect passivation due to imbalanced concentration of charge carriers (electrons and holes). However, due to the large contact resistance between c-Si(n)/LiF_x_, only a low V_OC_ value of 575 mV can be obtained [130]. It has also been suggested that a passivation layer is needed to mitigate this issue [106]. Bullock et al. [106] explored the contact structure of titanium oxide (TiO_x_) and a-Si:H(i) as a passivation layer, and obtained the best efficiency by using a-Si:H(i). A PCE of 19.42% was reported with a combination of a-Si:H(i)/LiF_x_/Al as an electron-selective contact and a-Si:H(i)/MoO_x_ as a hole-selective contact, representing a considerable increase (from 14% to 20%) in the efficiency of dopant-free asymmetric SHJ solar cells [106]. In another study, the interlayer of TiO_x_ (c-Si(n)/a-Si:H(i)/TiO_x_/LiF_x_) was found to exhibit thermal stability of contact resistivity [131]. In the absence of TiO_x_, the contact resistance increases after 150 °C, which is not conductive to carriers [20]. From the tests of the effect of annealing on the V_OC_, it was found that the cells with TiO_x_ showed almost no decrease in V_OC_. However, there was a significant decrease in V_OC_ in the samples without TiO_x_, which was likely due to the degraded interfacial passivation quality because of the intermixing between the a-Si:H(i) and LiF_x_/Al layers as the annealing temperature increased [131]. The same result was found in another study, where the insertion of TiO_x_ prevented the diffusion of Rb and F elements into the a-Si:H(i) layer, as shown in Figure 8 [132]. As such, the degradation of passivation quality was suppressed. TiO_x_, as a relatively low WF (~4 eV) of TMO, alone has been studied as an electron-selective material [133]. As shown in Figure 9a, the ultra-thin TiO_2_ (4.5 nm) film not only provides a good passivation effect on the silicon surface, but also enables low contact resistivity at the c-Si/TiO_2_ interface [134]. The insertion of the ultra-thin TiO_2_ film significantly improved the V_OC_ (from 585 mV to 638 mV) and FF (from 78.2% to 79.1%) of the cell. The formation of Si-O-Ti bonds at the c-Si/TiO_2_ interface is responsible for the excellent surface passivation provided by the ultrathin TiO_2_ films. High temperature annealing (>250 °C) caused the breakage of Si-O-Ti bonds [135] as well as the transition from the amorphous to the anatase phase [134], leading to a decrease in passivation quality. It was also demonstrated in another study that the formation of anatase phases induced by increased thickness (5 nm) and crystallization led to a decrease in passivation quality [135]. As shown in Figure 9b,c, the Schottky barrier (ϕ_B_) is reduced due to the insertion of TiO_2_ between c-Si(n) and Al [134]. This results in a small ∆E_C_ of approximately 0.05 eV, which favors electron transport through the TiO_2_ layer. On the other hand, a large ∆E_V_ of approximately 2.0 eV is created, which effectively blocks the transport of holes [136,137].

The primary advantage of using dopant-free carrier-selective materials over doped a-Si:H is their larger band gaps, which reduces optical absorption loss. This advantage is achieved while maintaining good passivation through the use of the SHJ structure. By applying dopant-free materials for carrier selective contact, the use of poisonous gases (for example, diborane and phosphine) for doping can be avoided in SHJ solar cells. At present, cells with dopant-free layers did not reach the same efficiency level as amorphous silicon-doped SHJ solar cells. Considering the potential of carrier selectivity and dopant-free benefits, it is worthwhile to further explore their potential for better performance on SHJ structures.

## 3. Summary of Recent Highlights in High-Efficiency SHJ Solar Cells

In this chapter, a discussion is provided on the developments of surface cleaning and high-efficiency SHJ solar cells within the last 5 years.

Hydrogen plasma etching (HPE) exhibits both chemical passivation and field-effect passivation [138]. As shown in Figure 10a, a τ_eff_ value of 6.1 ms was obtained by using HPE cleaning, which corresponds to an i-Voc of 750 mV, indicating 1.4 ms τ_eff_ or 5 mV i-V_OC_ higher than that of the HF sample. Jia et al. [138] concluded that the enhancement in τ_eff_ could not be entirely attributed to interface defect hydrogenation. The obvious band bending in the plasma sample shown in Figure 10b indicates that the HPE-induced field - effect passivation could be the key to a higher τ_eff_. When n-type substrates are cleaned using HPE, the donor-H complex undergoes decomposition after annealing, which causes the accumulation of H^−^. This leads to a negatively charged surface, resulting in upward band bending of c-Si. The current two primary cleaning methods are RCA and ozone-based wet-chemical cleaning [139]. Several investigations continue to rely on RCA cleaning due to its procedural stability and repeatability. Several PV manufacturers have turned to ozone-based solutions that have been found to provide comparable passivation quality to traditional RCA cleaning, while being less expensive and environmentally friendly [139]. The cleaning methods are summarized in Table 1 with the corresponding passivation techniques leading to high-efficiency SHJ solar cells.

As can be seen from Table 1, a-Si:H(i) is widely used to achieve chemical passivation due to its notable passivation properties. Passivation structures can be single-sided or double-sided, both of which are effective in achieving good passivation [149,150]. The less dense i1 layer allows hydrogen to penetrate to the interface and passivate defects, while the dense i2 layer has fewer defects and is capable of forming high-quality films. This stacked intrinsic passivation layer was shown to provide excellent passivation quality for high-efficiency SHJ solar cells, as demonstrated by efficiency of 24.51% with one-sided a-Si:H(i) film [143] and efficiency up to 25.11% with two-sided a-Si:H(i) film [144]. High-performance nanocrystalline materials have demonstrated good carrier selectivity in high-efficiency SHJ solar cells. Seed layers are also shown to be important to the carrier-selective layer to promote the nucleation of nanocrystalline materials. Qiu et al. [145] showed that the stacked structure of nc-Si:H(n) and nc-SiO_x_:H(n) outperformed single-layer nc-SiO_x_:H(n). As a seed layer, nc-Si:H(n) can increase the crystallinity of nc-SiO_x_:H(n), resulting in a small ∆E_C_ at the interface of nc-SiO_x_:H(n)/ITO, facilitating electron collection with an efficiency of 22.9% (J_SC_ of 38.6 mA/cm^2^, V_OC_ of 738 mV, FF of 80.3%). Samples using nc-SiO_x_:H(n)/nc-Si:H(n)/ITO resulted in the highest efficiency of 23.1%. This is due to the higher electron affinity of nc-Si:H(n) compared to nc-SiO_x_:H(n) [151,152], which aids in the collection of electrons into the ITO layer. It demonstrates that the electron barrier at the ESL/ITO interface can be lowered by inserting an nc-Si:H(n) layer above or below the nc-SiO_x_:H(n) layer, which is advantageous for electron transport and collection.

Compared to a single light-receiving side layer (window layer) [153], stacked n-type nc-SiC:H(n) can achieve a balance between passivation and conductivity [97]. For the single-layer µc-SiC:H(n), the optimal PCE of 21.6% (V_OC_ of 709 mV, J_SC_ of 39.5 mA/cm^2^ and FF of 77.1%) was achieved by using HWCVD at T_f_ of 1845 °C. In addition, the study also reported a higher V_OC_ at lower T_f_, which is due to lower concentration of hydrogen groups at lower T_f_, resulting in a lower degree of etching of the SiO_2_ passivation layer and a superior passivation quality [153]. The use of a stacked structure (Figure 11) further improves passivation and efficiency. For the first layer, 3 nm nc-SiC:H(n) was deposited at a lower T_f_ (1775 °C) to guarantee passivation quality, followed by a higher T_f_ of 1950 °C (30 nm) to guarantee conductivity, with a PCE up to 23.99% (V_OC_ of 725 mV, J_SC_ of 40.87 mA/cm^2^, and FF of 80.9%). In the condition of intrinsic and doped a-Si:H for passivation, it has been reported that increasing the doping efficiency of B-Si_4_ resulted in improved field-effect passivation and conductivity. Such findings can be attributed to the diffusion and hopping of bonded hydrogen atoms. As a result, a high PCE of 25.18% with a high FF of 85.42% was achieved [147]. Moreover, a higher PCE of 26.3% with V_OC_ up to 750 mV was validated by using microcrystalline material [142]. The optimization of the microcrystalline window layer was reported to be a significant reason for several breakthroughs in the efficiency of SHJ solar cells [142].

By attempting to utilize single-sided or double-sided dopant-free materials as carrier selective layers, a PCE over 20% was obtained [131,148,154,155]. Among such materials, a prominent efficiency of 23.5% was ensured by the MoO_x_ hole transport material [148]. The authors also maintained that the thickness of the intrinsic layer showed gentle impact on the FF of the cell, indicating that the use of MoO_x_ with a high WF provided strong hole carrier selectivity, reducing the dependency of FF on the thickness of the a-Si:H(i) layer. In their study, MoO_x_ exhibits a more slightly substoichiometric state, which can be corrected through deposition and annealing processes, resulting in a total oxidation state of Mo that is close to its initial +6 state [148]. This high oxidation state is indicative of a high WF [125], which contributes to the material’s greater hole selectivity. He et al. [132] investigated the combination of MoO_x_ hole-selective contact and RbF_x_/Al electron-selective contact to replace the doped a-Si:H layer in SHJ solar cells. By inserting a thin TiO_x_ interlayer between the a-Si:H(i) and RbF_x_/Al, the recombination at the electron-selective contact interface was well suppressed, leading to a PCE of 22.9% and a V_OC_ of 709 mV. Such research provides an effective way to further improve the efficiency and reduce the cost of SHJ solar cells in a dopant-free structure. Notably, for double-sided dopant-free materials, lower V_OC_ values are shown, indicating a future potential to enhance efficiency by improving interface passivation.

## 4. Conclusions

In this study, the surface cleaning methods and passivation technologies were explored and summarized. Ozone-based cleaning is not only capable of achieving high-quality passivation, leading to a high V_OC_ of 749 mV, but is also a cost-effective alternative to the RCA method. These qualities make it a trending technology for future surface cleaning in the photovoltaic field. The use of stacked passivation layer is anticipated to become a widespread practice in high-efficiency solar cells to improve both chemical and field-effect passivation, due to its ability to achieve a balance between conductivity and transparency. Additionally, nanocrystalline-alloyed materials with larger band gaps and higher doping efficiency have been found to outperform traditional doped a-Si:H materials, resulting in a high V_OC_ up to 750 mV. Dopant-free materials show considerable potential for carrier-selective contacts in SHJ solar cells. By summarizing the characteristics of high-efficiency solar cells, we aim to guide the readers to more advanced discoveries in surface cleaning and passivation technologies, which are beneficial for the development of high-efficiency SHJ solar cells.

## Figures and Tables

**Figure 1 materials-16-03144-f001:**
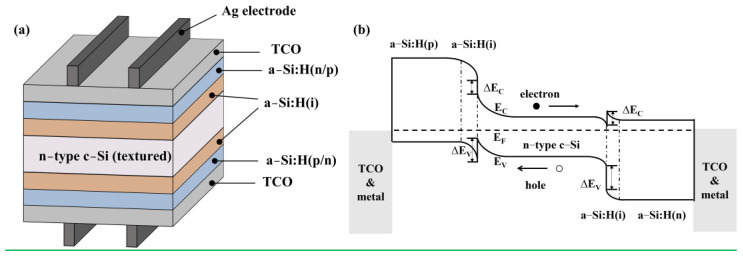
(**a**) A sketch of the structure of a SHJ solar cell made from the n-type c-Si substrate. (**b**) A schematic band diagram for heterojunction structure of n-type SHJ solar cell. The arrows point in the transport direction of electrons and holes. E_F_ denotes the Fermi energy level. ∆E_C_ and ∆E_V_ stand for conduction band offset and valence band offset, respectively. The band bending at the a-Si:H/TCO interfaces is not drawn (for simplicity). For the front emitter and rear emitter solar cells, light is incident from the left and right sides, respectively.

**Figure 2 materials-16-03144-f002:**
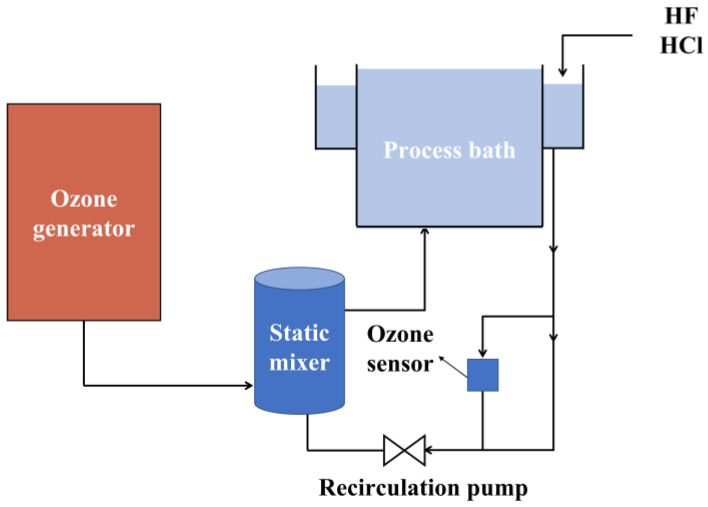
A schematic diagram of the ozone-based cleaning process.

**Figure 3 materials-16-03144-f003:**
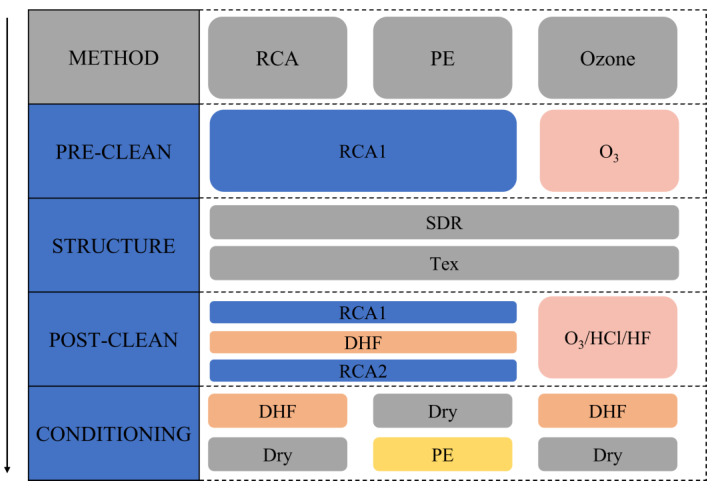
Comparison of three surface cleaning techniques for specific chemicals and processes. For a better understanding, all cleaning steps before deposition are shown. The direction of the arrow on the left indicates the cleaning sequence. The abbreviations of SDR and Tex denote saw damage removal and texturing. Plasma etching (PE) is a dry etching method to remove silicon oxide layer.

**Figure 4 materials-16-03144-f004:**
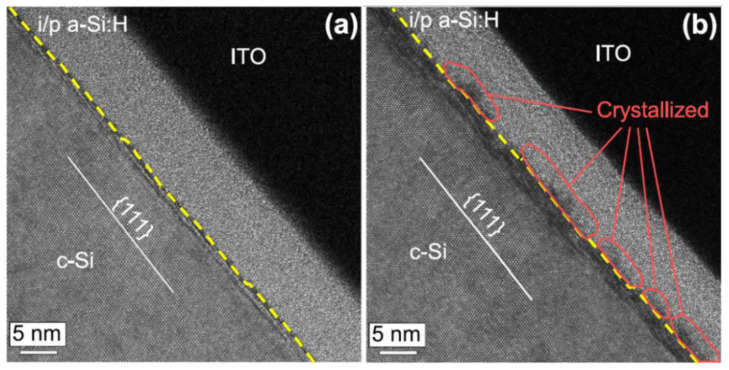
Cross-sectional TEM images of the interfaces between i/p a-Si:H and c-Si of the SHJ solar cells: (**a**) stack of less dense a-Si:H(i1) layer and dense a-Si:H(i2) layer, (**b**) single dense a-Si:H(i2) layer [75]. The crystallized interface indicates the epitaxial growth in the condition of a single dense a-Si:H(i2) layer.

**Figure 5 materials-16-03144-f005:**
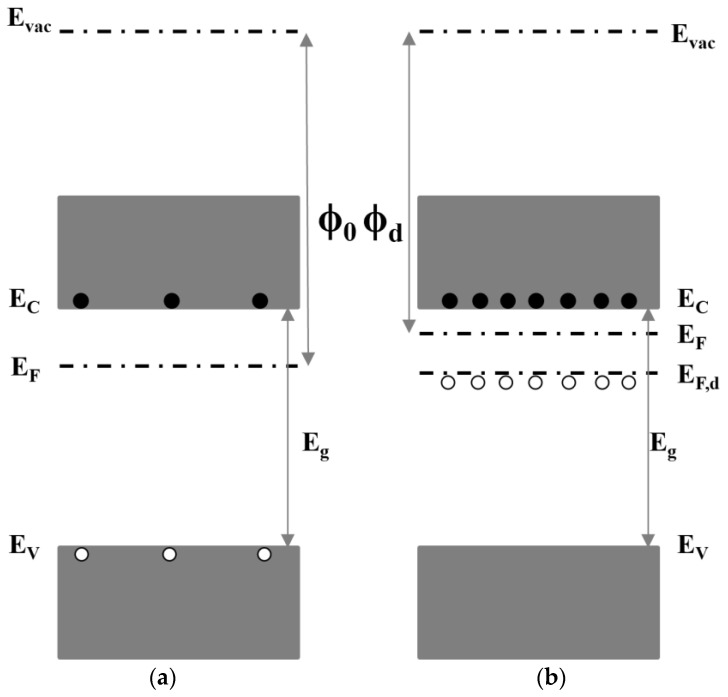
Schematic diagram of the energy level without (**a**) and with (**b**) oxygen deficiency in MoO_x_. With the introduction of oxygen deficiency, the work function of MoO_3_ (ϕ_0_) decreases to ϕ_d_.

**Figure 6 materials-16-03144-f006:**
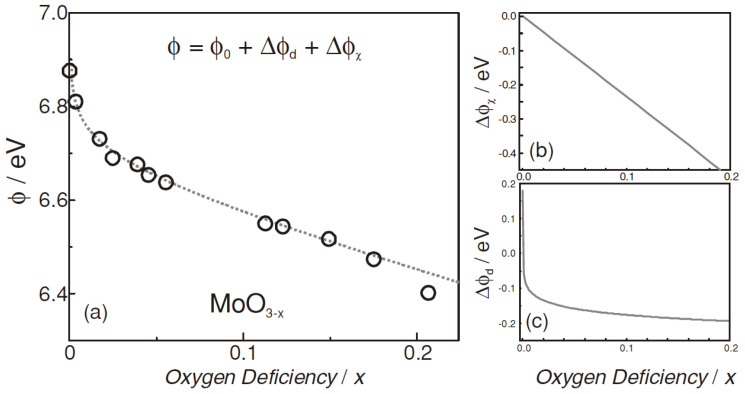
(**a**) Plot of oxygen deficiency (vacancy) and work function (ϕ). ϕ_0_ is the work function of MoO_3_, and the work function of MoO_x_ decreases with the increase of oxygen deficiency, which results from (**b**) low electronegative cation, causing work function change (Δϕ_χ_) and (**c**) donor state, causing work function change (Δϕ_d_) [125].

**Figure 7 materials-16-03144-f007:**
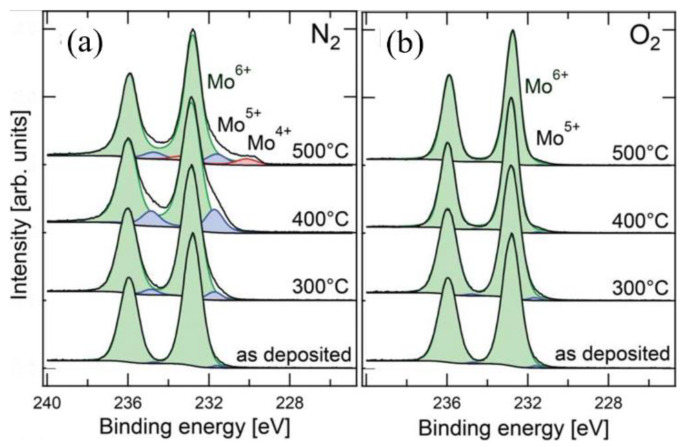
X-rays photoelectron spectroscopic (XPS) spectra of Mo 3d, showing the effect on annealing temperature and ambient: (**a**) N_2_, (**b**) O_2_. Reprinted with permission from [123]. Copyright 2014 American Chemical Society.

**Figure 8 materials-16-03144-f008:**
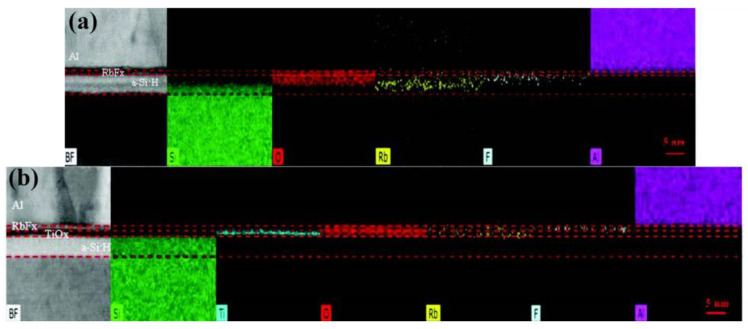
Cross-sectional energy dispersive X-ray spectroscopy (EDX) distributions without and with TiO_x_ layer: (**a**) In the absence of TiO_x_, there is observable penetration of O, Rb, and F, which results in degraded passivation quality. (**b**) There is no elemental penetration in the condition of the TiO_x_ interlayer [132].

**Figure 9 materials-16-03144-f009:**
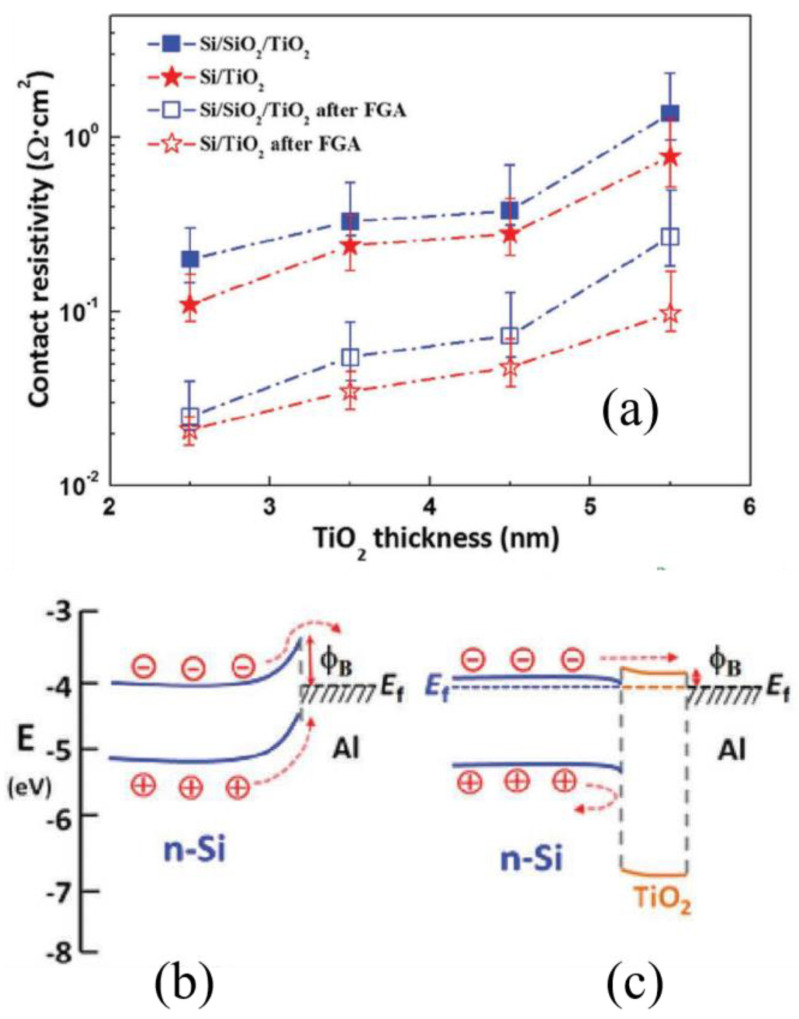
(**a**) The contact resistivity dependence on the thickness of the TiO_2_ layer before and after forming gas annealing (FGA). The Si/TiO_2_ interface exhibits good passivation and relatively low contact resistivity. (**b**) and (**c**) Band gap alignments with and without TiO_x_ interlayer between the silicon wafer and Al [134]. A minus sign in the circle represents the electron, and a plus sign represents the hole. With the insertion of the TiO_x_ layer, the decreased ϕ_B_ could favor the electrons to be collected and holes to be blocked, showing the electron-selective property of TiO_2_.

**Figure 10 materials-16-03144-f010:**
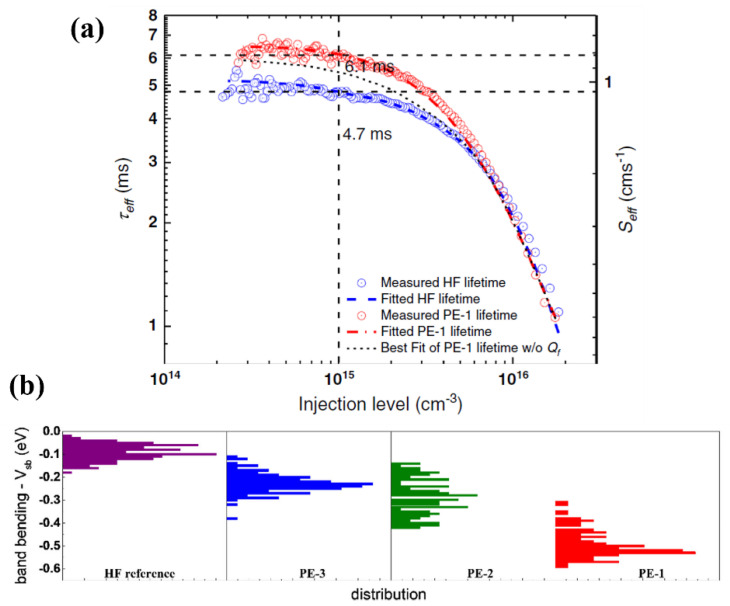
(**a**) The τ_eff_ and the corresponding effective surface recombination rate (S_eff_) of HF and PE-1 samples as a function of injection level. (**b**) The band bending of the HF and PE samples [138]. The difference between the PE samples is the hydrogen flow rate and pressure, all other parameters are constant.

**Figure 11 materials-16-03144-f011:**
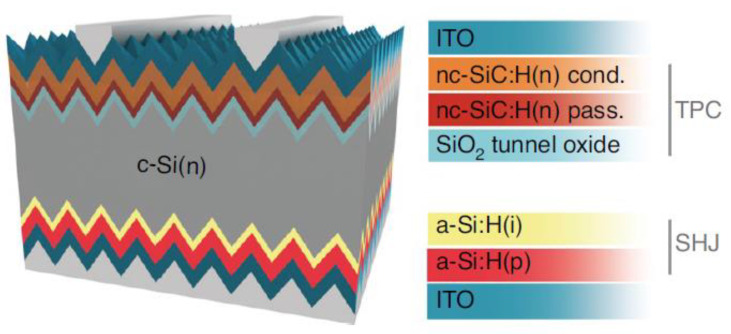
Schematic structure of a highly transparent passivating contact (TPC) provided by the stacked nc-SiC:H(n) window layer [97].

**Table 1 materials-16-03144-t001:** Performance, contact features and cleaning methods of high-efficiency SHJ solar cells.

Features	Cleaning	PCE [%]	FF [%]	V_OC_ [mV]/ δ [µm]	J_SC_ [mA/cm^2^]/ Area [cm^2^]	Reference/ Year
SHJ-IBC, a-Si:H contact	-	26.7	84.9	738/ 165	42.65/ 79	[140,141]/ 2017
Microcrystalline layer, new i layer	-	26.3	86.59	750.2/ -	40.49/ 274.3	[142]/ 2022
SHJ	-	25.1	83.5	738/ 160	40.8/ 151.9	[8]/ 2015
a-Si:H(i1)/a-Si:H(i2)/a-Si:H(p) at backside	Ozone	24.51	83.61	741.8/ 180	39.52/ 244.53	[143]/ 2022
Both sides i1 + i2 a-Si:H a-Si:H(n)/nc-SiO_x_:H(n) as front ESL	-	25.11	84.98	747/ 150	39.55/ 244.45	[144]/ 2020
nc-SiOx:H(n)/nc-Si:H(n) as front ESL	Ozone	23.1	80.7	739/ 170	39.0/ 244	[145]/ 2020
Stacked nc-SiC:H(n) as front ESL, SiO_2_ tunnel oxide at front side	RCA	23.99	80.9	725/ 170	40.87/ 3.487	[97]/ 2021
(Seed layer) nc-Si:H(n)/nc-SiO_x_:H(n) as front ESL	RCA	24.09	84.5	744/ 140	38.3/ 243.36	[146]/ 2022
Enhanced doping efficiency of B-Si_4_	RCA	25.18	85.42	749.1/ 160	39.36/ 244.63	[147]/ 2022
MoO_x_ as front HSL, a-Si:H(n)/µc-Si:H(n) as back ESL	-	23.5	81.8	734/ 180	39.2/ 3.972	[148]/ 2020
MoO_x_ as back HSL a-Si:H(i)/TiO_x_/RbF_x_/Al as front ESL	RCA	22.9	79.6	709/ 145	40.5/ 2.56	[132]/ 2022
MoO_x_ as front HSL a-Si:H(i)/TiOx/LiF_x_/Al as back ESL		20.7	76.2	706 250	38.4 4	[131] 2018

The features of a SHJ solar cell are responsible for both high-quality passivation and its difference from the conventional contact structure of a-Si:H(n)/a-Si:H(i)/c-Si(n)/a-Si:H(i)/a-Si:H(p). The first three cells are included as reference and consist of an IBC-SHJ cell, SHJ cell with microcrystalline window layer, and SHJ cell with conventional amorphous silicon contacts. δ represents the wafer thickness. “-” indicates that the corresponding information has not been disclosed.

## Data Availability

Not applicable.

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
