# Peer review of "Surface Cleaning and Passivation Technologies for the Fabrication of High-Efficiency Silicon Heterojunction Solar Cells"

_materials, 2023, doi:10.3390/ma16083144_

Round 1
Reviewer 1 Report
In this work Shi et al reviewed the methodologies for surface passivation and cleaning when fabricating SHJ solar cells. The review is well written and comprehensive and could represent a useful guide for scientists interested in improving the efficiency of solar cells by physicochemical treatments. There are, however, several issues that should be clarified before a recommendation of acceptance could be made as noted below:
a) As this is a review, it is the responsibility of the authors to indicate in the manuscript whether the field, or closely related fields, have already been recently reviewed and, if so, to discuss the differences with previous reviews.
b) There is no sufficient attention to the chemical processes in this review that, as it is, is too much descriptive. Chemical processes should be drawn rather than described. For instance at Page 8 the authors wrote "....at higher filament temperature (Tf) results in a decrease in i-VOC, which is due to increased hydrogen group density....". Of course, this makes no sense in chemistry (hydrogen group density?). In general, the authors should had their own schemes to describe the occurring chemical processes.
c) In the conclusions the authors should more clearly indicate which chemical cleaning and passivation technology they consider more efficient and worth of investigation in the future and they should justify such proposal.
Reviewer 2 Report
The authors have made a great contribution to the field of material science and submitted a very interesting and well-presented experimental result in the manuscript entitled "Surface cleaning and passivation technologies for the fabrication of high-efficiency silicon heterojunction solar cells". The reported experimental results are comprehensive, complementary justifying the various discussions within the text and drawing interesting conclusions. The present manuscript is suitable for publication and lies under the scope of this journal (Materials). However, there are some drawbacks and questions which must be answered before the final publication in the journal Materials:
1. Fabrication process of SHJ is not properly explained (describe the method properly).
2. Novelty of this work should be explained in the introduction part.
3. Inappropriate references are cited in the manuscript. It can be reduced.
Reviewer 3 Report
This review is devoted to a topical issue - a way to increase solar cells based on silicon heterostructures. This review covers the main publications on this topic in recent years. It will be useful and interesting to a wide range of readers.
However, there are a number of small comments to the text of the review:
1. Authors should avoid abbreviations and abbreviations coinciding with chemical formulas, such as DIO3 (line 131 onwards)
2. It is better not to use the term "ammonium hydroxide, NH4OH" (lines 104-105). From a chemical point of view, " aqueous ammonia solution, NH3 ∙ H2O" is more correct
3. The authors use the notation teff (line 150 et seq), but do not give its decoding in the text of the article. Apparently, this means the lifetime of photogenerated current carriers. It needs to be clarified. If this is so, then these times also depend on the light intensity, and it is not quite correct to compare the values from different works, where, probably, these intensities differed.
4. The authors also use the value Tf (line 545 et seq.). From the context it is clear that this is some kind of temperature, however, the following values more higher than the melting point of crystalline silicon. What is meant here?
5. In the text of the article, Voc is expressed either in mV or in V. The values should be agreed upon.
Reviewer 4 Report
In this manuscript, shi et al. reviewed the surface cleaning and passivation technologies for silicon heterojunction solar cells. The authors need to further clarify this manuscript, here below are my suggestions.
1. Please explain clearly in the main text, why there is a need for this review article, especially with this topic.
2. The majority of the references cited in the introduction part are too old. Therefore, please review the latest developments in the field.
3. Please provide the summary and perspectives for this review article.
4. Please provide your suggestions on the topic and how it is useful for future developments.
5. English needs to be polished further.

Round 2
Reviewer 1 Report
I believe the authors have provided an acceptable version of their review although I would have preferred to see more chemical schemes to explain the chemical processes.
Reviewer 4 Report
The authors have answered the queries, thus I am supportive of its publication in its current form.